# Molecular Mechanism and Agricultural Application of the NifA–NifL System for Nitrogen Fixation

**DOI:** 10.3390/ijms24020907

**Published:** 2023-01-04

**Authors:** Wenyao Zhang, Yihang Chen, Keyang Huang, Feng Wang, Ziqing Mei

**Affiliations:** 1Key Laboratory of Molecular Medicine and Biotherapy, School of Life Science, Beijing Institute of Technology, Beijing 100081, China; 2College of Life Sciences, Yan’an University, Yan’an 716000, China; 3School of Chemistry and Biological Engineering, University of Science and Technology Beijing, Beijing 100083, China

**Keywords:** biological nitrogen fixation, nitrogenase, NifA–NifL system, biological nitrogen fertilizer, agricultural application

## Abstract

Nitrogen–fixing bacteria execute biological nitrogen fixation through nitrogenase, converting inert dinitrogen (N_2_) in the atmosphere into bioavailable nitrogen. Elaborating the molecular mechanisms of orderly and efficient biological nitrogen fixation and applying them to agricultural production can alleviate the “nitrogen problem”. *Azotobacter vinelandii* is a well–established model bacterium for studying nitrogen fixation, utilizing nitrogenase encoded by the *nif* gene cluster to fix nitrogen. In *Azotobacter vinelandii*, the NifA–NifL system fine–tunes the *nif* gene cluster transcription by sensing the redox signals and energy status, then modulating nitrogen fixation. In this manuscript, we investigate the transcriptional regulation mechanism of the *nif* gene in autogenous nitrogen–fixing bacteria. We discuss how autogenous nitrogen fixation can better be integrated into agriculture, providing preliminary comprehensive data for the study of autogenous nitrogen–fixing regulation.

## 1. Introduction

Nitrogen is the core component of biological molecules, such as proteins and nucleic acids. Nitrogen makes up as much as 78% of the atmosphere, but it is not utilized directly by most creatures and can be effectively absorbed only when converted into ammonia or ammonium salts. An increase in available nitrogen content can significantly improve the yield of crops in poor soils [1]. At present, the best way to increase the available nitrogen content of crops is by applying chemical nitrogen fertilizer. However, there are two major disadvantages: First, the utilization rate of nitrogen fertilizer is extremely low, and excessive application of nitrogen fertilizer leads to water eutrophication and causes an imbalance in the species distribution in the water ecosystem, thus resulting in the gradual extinction of the whole water ecosystem. Second, for many smallholders in some developing countries, such as Sub–Saharan Africa, the scarce availability and high cost of nitrogen fertilizer make it unusable; therefore, these smallholders suffer from low yields [2,3,4,5]. Finding clean alternatives to nitrogen fertilizers is essential for sustainable and safe agricultural development.

The vast majority of nitrogen fixation is executed by nitrogen–fixing microorganisms [6]. Biological nitrogen fixation is the process in which nitrogen–fixing microorganisms use nitrogenase to directly reduce atmospheric nitrogen to ammonia [7]. This process introducing nitrogen–fixing bacteria or nitrogen–fixing enzymes to the crop provides an opportunity to increase the available nitrogen content of the crop and improve crop nutrition.

Biofertilizers are cheaper, require less capital to use, and are thus increasingly important to agriculture. Biofertilizers, also known as biological inoculants, are organic preparations containing microorganisms [8]. When used as a seed treatment or when seedling roots are immersed in seeds or soil fertilization, they proliferate rapidly and form dense populations in the rhizosphere that rapidly fix nitrogen, increasing the available nitrogen content of the crop. Beneficial biofertilizers for crop production contain nitrogen–fixing bacteria, azospirals, cyanobacteria, green algae, etc. [9,10,11]. However, the application of most nitrogen–fixing microorganisms as biological nitrogen fertilizers in agricultural production faces various obstacles, such as harsh growth conditions and low nitrogen–fixation efficiency. Overcoming these obstacles with biotechnology will facilitate the wide application of biological nitrogen fertilizers in agricultural production, increasing the available nitrogen content of the crop for large agricultural returns.

According to the characteristics of nitrogen fixation, nitrogen–fixation microorganisms can be divided into three types: autogenous nitrogen–fixation bacteria, symbiotic nitrogen–fixation bacteria, and combined nitrogen–fixation bacteria [12]. Autogenous nitrogen–fixing bacteria are free–living bacteria that can fix nitrogen. *Azotobacter vinelandii* (*A. vinelandii*), with its high expression and nitrogenase activity, can perform efficient nitrogen fixation under aerobic conditions and is thus becoming a model bacterium for the study of autogenous nitrogen–fixing bacteria [13]. *A. vinelandii* contains three types of nitrogenases, all of which are multi–subunit protein complexes that bind metal ions, namely, ferric–molybdenum (Fe–Mo) nitrogenase, ferric–vanadium (Fe–V) nitrogenase, and ferric–ferric (Fe–Fe) nitrogenase [14]. The nitrogen–fixing reactions are mainly catalyzed by Fe–Mo nitrogenase, whose expression is regulated by a NifA–NifL system. In this manuscript, we summarized the molecular mechanisms of the regulation of the Fe–Mo nitrogenase expression via NifA–NifL and their application in agricultural development, supplying preliminary comprehensive data for the study of autogenous nitrogen–fixation regulation.

## 2. Nitrogenase and Its Transcriptional Regulation

### 2.1. Nitrogenase

Nitrogenase is a complex oxygen–sensitive metalloenzyme with three isoforms: Fe–Mo nitrogenase (encoded by *nif* gene cluster), Fe–V nitrogenase (encoded by *vnf* gene cluster), and Fe–Fe nitrogenase (encoded by *anf* gene cluster) (Figure 1a). All known diazotrophs contain at least one of the three closely related nitrogenase isoforms. Although they have different metal contents, these nitrogenase isoforms are related to each other in terms of their structure, mechanism of action, and phylogeny [15].

The alignment of protein sequences showed that the three nitrogenases share a similar basic structure. Among them, Mo–Fe nitrogenase plays a dominant role in nitrogen fixation, while Fe–V nitrogenase and Fe–Fe nitrogenase act as alternative nitrogenases. Alternative nitrogenases are activated only if the molybdenum in the environment is insufficient. Of these, the most widely studied subtype is Mo–Fe nitrogenase, which exhibits the highest nitrogen–fixation efficiency. The catalytic center of Mo–Fe nitrogenase, which is composed of a Mo–Fe protein (named dinitrogenase) and Fe protein (named nitrogenase reductase), is encoded by *nif*H, *nif*D, and *nif*K. The Fe protein is a γ2–type homodimer encoded by the *nif*H gene, and the Mo–Fe protein is an α2β2–type heterotetramer in which the *nif*K gene encodes an α subunit and the *nif*D gene encodes a β subunit. Both ends of the Mo–Fe protein can bind the Fe protein. Recent studies have reported that only the single–headed complex assembled by one molecule of a Mo–Fe protein and one molecule of an Fe protein is required for nitrogen fixation, and the double–headed complex exists in a trapped state [16]. Mo–Fe nitrogenase has a unique Mo–Fe cofactor located at the active site of Mo–Fe protein, which participates in the nitrogen–fixing reaction [15]. During nitrogen fixation, Fe protein couples and hydrolyzes two ATP molecules, while one electron is transferred to the 8Fe–7S cluster in the Fe protein by the 4Fe–4S cluster. Afterward, the electrons are transferred to the Mo–Fe cofactor to participate in the reduction of the N_2_ molecule. N_2_ forms a stable intramolecular triple bond so the reduction of N_2_ to NH_3_ by nitrogenase is an extremely difficult reaction that requires overcoming the activation energy barrier of catalysis (Figure 1b) [17]. High–energy electrons stored in the Mo–Fe cofactor can be used as reducing agents to overcome the activation energy barrier of catalysis in the presence of large amounts of ATP and protons [16]. This catalytic mechanism is also applicable to V–Fe nitrogenase and Fe–Fe nitrogenase, but they need more high–energy electrons, ATP, and protons to overcome the energy barrier. Mo–Fe nitrogenase consumes at least 16 ATP molecules to immobilize a molecule of N_2_, whereas V–Fe nitrogenase and Fe–Fe nitrogenase consume 24 to 32 ATP molecules (Figure 1c) [18]. The conversion efficiency of nitrogenase is extra low due to the complexity of the reaction catalyzed by nitrogenase. In order to satisfy the growth requirements, nitrogen–fixing bacteria express a large amount of nitrogenase, which generally accounts for 10–20% of the total protein content [13].

V–Fe nitrogenase and Fe–Fe nitrogenase are encoded by *vnf*HDGK genes and *anf*HKDG genes, respectively; whereas the *vnf*H (*anf*H) gene encodes nitrogen–fixing reductase; and *vnf*KDG (*anf*KDG) genes account for the synthesis of the α subunit, β subunit, and δ subunit to form the α2β2δ2 hexamer (Figure 1b). Although the specific function of the δ subunit encoded by *vnf*G (*anf*G) remains unclear, it is essential for maintaining nitrogenase activity. The nitrogen–fixing mechanism of V–Fe nitrogenase and Fe–Fe nitrogenase is similar to that of Mo–Fe nitrogenase. In regard to the development and evolution of nitrogenase, some views have proposed that V–Fe nitrogenase and Fe–Fe nitrogenase might have evolved from Mo–Fe nitrogenase [19].

### 2.2. Transcriptional Regulation of Nitrogenase

Nitrogenase is extremely sensitive to oxygen, and its catalytic reduction of nitrogen molecules to ammonia must be carried out in a strictly anaerobic microenvironment. As result, the expression of nitrogenase requires a specific microenvironment, where strict anaerobic activity is necessary to maintain the activity of nitrogenase. Additionally, the cell should be in the peak metabolic stage with a large amount of ATP and sufficient nitrogen to ensure the precise modulation of nitrogenase expression by nitrogen–fixing bacteria. The biosynthesis of active nitrogenases relies on a variety of Nif proteins encoded by *nif* genes beyond the structural subunits of the catalytic center, including the molecular scaffold protein gene, the metal cluster carrier protein gene, and the metal cofactor biosynthesis gene. It was reported that at least nine nif genes are required for the synthesis of bioactive Mo–Fe nitrogenases: *nif*H, *nif*E, *nif*N, *nif*S, *nif*U, *nif*V, *nif*Y, *nif*B and *nif*Q, which performs functions including redox provisioning and electron transport [20].

The transcriptional regulation of nitrogenase might differ between nitrogen–fixing bacteria. For instance, *A. vinelandii* displays an expression and transcriptional regulation system of Mo–Fe nitrogenase, encoded by the *nif* cluster, which includes a major *nif* cluster and a minor *nif* cluster. The major *nif* cluster contains five major gene clusters (*nif*HDKTY, *nif*ENX, orf5, iscA*^nif^nif*USV–cysE1*nif*–*nif*WZM–clpX2, and *nif*F) encoding the nitrogenase complex. The *nif*HDKTY gene cluster contains the structural genes of Mo–Fe nitrogenase, which include *nif*H, *nif*D, *nif*K, *nif*T, and *nif*Y. Moreover, the *nif*LA operon, located at the minor nif cluster (*rnf*ABCDGEH, *nif*LAB, *fdx*N, *nif*OQ, rhdN, and grx5^nif^), encodes the NifA protein and NifL protein that regulate and control the transcription of nitrogenase (Figure 1a) [21]. Since the first discovery of the function of the *ni*fLA operon, the research on the NifL–NifA binary regulatory system has achieved significant progress: the function of nitrogen fixation is regulated by the *nif*LA operon in *A. vinelandii*, *nif*A distal, and *nif*L proximal to the promoter (Figure 1a) [8]. Moreover, the expression of the *nif*LA gene is rarely affected by environmental factors, and it is continuously expressed or has little change in various growth stages [21,22]. NifA is the transcriptional activator protein of the *nif* gene cluster, while NifL inhibits the activity of NifA via interacting with NifA. The NifL–NifA system is affected by the intracellular redox environment and binding state of the ligand (2–OG, ATP/ADP, FAD). The reversible modification of GlnK by uridylation also has a regulatory effect on the transcription of the NifL–NifA system. When there is excess nitrogen, nitrogen metabolism results in low concentrations of 2–OG and high concentrations of glutamine. Glnk, NifA, and NifL can form the ternary complex to suppress the activity of NifA, which results in the blockage of nitrogenase expression. When nitrogen is limited, the floundering nitrogen metabolism results in the accumulation of 2–OG and the excessive consumption of glutamine, which disrupts the formation of the ternary complex to activate NifA. The active NifA can promote the expression of nitrogenase (Figure 2).

## 3. The Function and Structure of NifA

NifA is a bacteria enhancer binding protein (bEBP) containing AAA+ domain proteins, which can use the energy released by the hydrolysis of ATP to catalyze the conformational change of the σ54 factor, followed by transition of the σ54–RNAP holoenzyme from the inactive state to the active state, thus activating the transcription of the *nif* gene cluster [23]. Based on their biological function and regulatory patterns, bEBPs are classified into five categories. NifA is grouped in the third category, which contains the N–terminal GAF domain, central AAA+ domain, and C–terminal HTH domain (Figure 2a). The GAF domain is a regulatory domain widely found in many proteins, which can sense and transmit a variety of metabolic signaling molecules, such as cAMP, cGMP, glutamic acid, α–ketoglutaric acid (2–OG), and porphyrin ring [24]. The binding of 2–OG to the GAF domain activates NifA in an unknown manner and initiates the transcription of the *nif* gene cluster in *A*. *vinelandii*. The AAA+ domain, a representative structural element of members of the AAA+ superfamily, can convert chemical energy into mechanical energy by binding and hydrolyzing ATP. Different from classic AAA+ clade 3 subfamily proteins, the AAA+ domain of NifA possesses two loop structures named L1 (GAFTGA motif) and L2 [25], which can interact with σ54 factor and reconstruct its conformation via hydrolyzing ATP [26]. The C–terminus of NifA is a helix–turn–helix motif of the DNA binding domain, which binds to upstream activator sequences (UAS) of the *nif* gene cluster promoter. Upon bending and cyclizing the DNA strand of the *nif* gene cluster, NifA bound to the UAS sequence can approach and interact with the σ54–RNAP holoenzyme in the promoter region, inducing the deformation of σ54 and activating the transcriptional activity of the RNAP–σ54 complex (Figure 3).

Despite its key role in the transcriptional activation of the *nif* gene cluster, a three–dimensional structure of NifA has not been reported. Fortunately, the partial structure of its homologous protein has been resolved, which improves the elaboration of the mechanism of transcriptional activation of NifA at large. NifA–like homolog 1 (Nlh1) is speculated to function as a σ54 activator in modulating the transcriptional activity of genes involved in nitrogen assimilation instead of nitrogen fixation [27]. The crystal diffraction and SAXS data of GAF domain revealed that the GAF domain forms a “back–to–back” dimer, and the dimeric coiled–coil signaling helix that connects the GAF domain with the ATPase domains repressed the assembly of the AAA+ domain by holding them in an inactive face–to–face orientation. The attachment of an unknown ligand to the ligand binding pocket of the GAF domain can activate Nlh1 by disrupting the dimeric coiled–coil signaling helix to release the ATPase domains for assembly [27,28]. NifA–like homolog 2 (Nlh2) is reported to act as an NtrC2 homolog in *Aquifex aeolicus*, which contains an N–terminal GAF domain rather than the receiver domain, similar to that of the NifA homolog. Like Nlh1, Nlh2 is also able to stimulate σ54–mediated transcriptional initiation. Studies have shown that the active activator forms a ring oligomer to facilitate the activation of the σ54–RNAP holoenzyme, such as a hexamer and heptamer. The phage shock protein F (Pspf) protein, a bEBP, is the only protein reported to form a complex structure with the σ54–RNAP holoenzyme. Recently, the Cryo–EM structure of the complex between the active PspF (hexamer) and σ54–RNAP holoenzyme was also determined, but the ATP binding site, the interaction sites between PspF and σ54, as well as the interface residues involving a polymer formation, require elucidation [23]. The AAA+ domain of the PspF protein from *Escherichia coli* (*E*. *coli*) exhibits a 48% similar sequence identity to that of NifA, suggesting that NifA is most likely involved in activating the σ54–RNAP holoenzyme in the form of a hexamer ring.

## 4. Structural Characteristics and Functions of the NifL Protein

As an anti–activator, NifL suppresses the transcription of nitrogenase gene activated by its partner protein NifA in *A. vinelandii* [29]. It is noteworthy that a specific molar ratio between NifL and NifA is required for this function, indicating that NifL directly interacts with NifA rather than modifying it. The complex of NifA and NifL has been found in a variety of microorganisms, including *A. vinelandii* and *Klebsiella pneumoniae* (*K. pneumoniae*) [30,31]. NifL suppresses the transcriptional activation of the nitrogenase gene via prohibiting the interaction of NifA with DNA and the complex formation of RNAP–σ54–DNA with NifA [31].

The NifL protein consists of four domains: the N–terminus contains two PAS (Per–ARNT–Sim) domains, named PAS1 and PAS2 [32] (shown in Figure 2a). The central region consists of a glutamine–rich superhelical structure containing a highly conserved histidine residue, referred as the H–domain; the C–terminal domain contains an ATP/ADP binding site composed of four highly conserved motifs (N, G1, F, and G2 motif), which is called the GHKL domain. Of these, the PAS domain is homologous to the GAF domain and can sense various environmental signals (such as oxygen, redox signaling, and light) via its α/β fold in response to different cofactors [33,34]. Both PAS1 and PAS2 are necessary for redox signal transduction corresponding to oxygen. The PAS1 domain of NifL binds to the flavin adenine di–nucleotide (FAD) cofactor, while the PAS2 domain can receive the redox signals from PAS1 and transmit them to the C–terminal H and GHKL domain [35]. The PAS2 domain is also identified in the NifL sequences of other aerobic diazotrophs [36]. The GHKL domain and H of NifL and the catalytic domain of histidine protein kinase (H ATPase–C) are homologous to the ATP binding domain of ATPases from the GHKL superfamily, but they are only capable of binding ATP/ADP rather than hydrolyzing ATP [33].

Until now, the detailed inhibition mechanism of NifA activity by NifL has remained elusive, but several studies have provided a basis for elucidating this mechanism. In vitro biochemical data proved that deletion of the GAF domain has no significant impact on the transcriptional activation of the nitrogenase gene. However, the deleted mutant of NifA required higher concentrations of NifL to inhibit its activity compared with the wild type [37], implying that the GAF domain of NifA is essential for the formation of the NifL–NifA complex. The activity of the NifA–E356K mutant, a single residue mutant in the AAA domain, is constitutive and insensitive to NifL, consistent with the fact that the NifA–E356K mutant can show higher activity than the wild type under anaerobic nitrogen–limiting conditions [38]. Another mutant, NifA–Y254N, can show resistance to NifL under anaerobic nitrogen–excessive conditions, but NifL is sensitive to the mutant under aerobic growth conditions [39,40]. These suggest that the AAA domain of NifA is also involved in the formation of the NifL–NifA complex. Thus far, no data have reported that the HTH domain of NifA is related to the formation of the NifL–NifA complex.

The surfaces of NifL that are required for the interaction with NifA are not well defined, but some experiments have established the fact that the region located between residues 287 and 360 corresponding to the H–domain might provide the main surface of the NifL–NifA complex. In vitro mutant assays proved that the NifL mutant with N–terminal PAS1 domain deletion (147~519 residues) can still form the NifL–NifA complex but not sense the redox signal [31]. This suggests that the PAS1 domain is not essential for the interaction between NifL and NifA. However, biochemical data regarding the mutant with the deletion of the H domain and GHKL domain showed that both of them were directly involved in the interaction with NifA [31]. The mutant with the replacement of arginine by cysteine at position 306 (NifL–R306C) can constitutively inhibit NifA, which further supports the conclusion that the H–domain provides the surface of the NifL–NifA complex [38]. There is no direct information showing that the PAS2 domain is involved in the interaction with NifA. Some protein structures with the PAS domain show that the conformational changes caused by signal sensation initiate the relay of the signal to effector domains. Oxidation of the FAD moiety results in a quaternary structure change in PAS1, resulting in a movement of the PAS2 protomers that is proposed to trigger rearrangement of the H and GHKL domains, promoting access to NifA and the formation of a NifL–NifA complex that inhibits its activity [33,35].

## 5. The NifL–NifA System Responds to the Transcriptional Regulation of Nitrogenase via Environmental Signaling Molecules

The NifL–NifA system of *A. vinelandii* integrates the intracellular redox and nitrogen and carbon status to regulate the expression of nitrogenase. The interaction between NifL and NifA is regulated in response to the intracellular redox environmental, ligand (2–OG, ATP/ADP, FAD/FADH_2_) binding status, and the signal–transduction protein GlnK. Under an adverse redox state (excess oxygen) or nitrogen–excess condition, oxidized NifL and NifA form binary complexes to suppress NifA activity. In addition, non–covalently modified GlnK can also interact with NifL to promote the formation of a GlnK–NifL–NifA ternary complex and inhibit NifA activity (Figure 4a). Relatively, in nitrogen–limiting conditions, 2–OG at a high concentration binds to the GAF domain and leads to uridylation of Glnk by GlnD. This can ensure NifA dissociation from the NifL–NifA complex so that free NifA can activate the transcription of the nitrogenase gene (Figure 4b) [29].

### 5.1. Regulation of NifA Function by 2–OG

As an important intermediate product of the tricarboxylic acid cycle (TCA), 2–OG is considered to be a key signal, which reflects the carbon metabolism status of cells. At the same time, 2–OG also provides the carbon skeleton for nitrogen assimilation, and its concentration indirectly corresponds to the status of intracellular nitrogen [41]. In vivo experiments indicated that the physiological concentration of 2–OG increases sharply from about 100 μmol/L to about 1 mmol/L, when the growth condition is changed from a nitrogen–excess state to a nitrogen–limiting state in *E*. *coli* [42]. In *A. vinelandii*, 2–OG directly affects the formation of the NifL–NifA complex in a concentration–dependent manner [36].

The binding of 2–OG to the GAF domain of NifA can regulate the response of NifA to NifL. The isothermal titration calorimetry (ITC) results showed that both the full–length NifA protein and GAF domain alone could bind 2–OG, and the affinity for either one of them is almost 60 μmol/L. The deletion of the GAF domain loses the ability to bind to 2–OG [43]. Limited protease hydrolysis experiments showed that 2–OG bound to the GAF domain increases the sensitivity of the GAF domain to trypsin digestion and inhibits the protection of these digestion sites by NifL [43]. This suggests that the binding of 2–OG probably leads to the allosteric reaction of the GAF domain, interrupting the inhibition of NifA by NifL. Consistently, the NifA–F119S mutant in the GAF domain is observed to lose the ability to bind with 2–OG without affecting the ability to bind to NifL, whereas the complex formed by NifA–F119S and NifL is no longer in control of 2–OG [36]. These results indicate that the binding of 2–OG to the GAF domain in a concentration–dependent manner induces the conformational changes in the GAF domain, which is followed by dissociation of the NifL–NifA complex, releasing the activity of NifA to activate transcription of the nitrogenase gene and promoting nitrogen fixation.

### 5.2. Effects of ADP and FAD Molecules on NifL Function

The formation of the NifL–NifA complex is associated with ATP/ADP and FAD/FADH_2_. The results of affinity chromatography proved that NifA forms a stable complex with NifL at a 1 mmol/L concentration of ADP, whereas the removal of ADP results in complex dissociation [44], suggesting a key role in reinforcing the stability of the NifL–NifA complex. ADP stabilizes the NifL–NifA complex by binding to the GHKL domain of NifL, enhancing its inhibitory activity [33,34,36]. The binding of 2–OG to the GAF domain of NifA alters the NifA conformation to antagonize the inhibitory activity of the ADP–bound NifL [36]. The catalysis of open promoter complexes by NifA requires hydrolysis of nucleotide triphosphate to supply energy, which is usually provided in the form of ATP or GTP. In vitro transcriptional experiments of open promoter complexes showed that ATP or GTP at a saturating concentration, with 4 mM GTP or 3.5 mM of ATP, can improve the formation of the inhibitory NifL–NifA complex, and the extra–low concentration of ADP (50 μM) can increase inhibition [45]. These data support the view that the accumulation of ADP promotes the formation of the NifA–NifL complex during nitrogen fixation, thereby regulating the efficiency of nitrogen fixation.

In the NifL–NifA system, FAD and FADH_2_ are two signal molecules that manipulate NifA activity via binding to NifL. FAD and FADH_2_ can reflect the intracellular oxidation/reduction status. The PAS1 domain of NifL can sense the redox status of FAD/FADH_2_ molecules. In the oxygenated state, the conformational change of NifL produced by the binding of the PAS1 domain to FAD prohibits the activity of NifA upon interacting with it. In addition, NifL resembles the oxidized NifL–NifA complex in the FAD spectral characteristics, regardless of the ADP binding state of NifL. This suggests that the FAD signal rather than ADP determines the inhibition of NifA activity by NifL [45]. Relatively, in the reduced state, the binding of the PAS1 domain to FADH_2_ causes NifL to abolish its ability to interact with NifA [46]. Previous studies have reported the crystal structure of the PAS1 domain bound to FAD. The structure showed that the PAS1 domain exists as a dimer in an asymmetric unit, and a novel cavity is formed inside each monomer, which can interact with FAD through salt–bridge, hydrogen–bond, and hydrophobic interactions. This structure supports the idea that hydrogen peroxide released by the oxidizing reaction of FAD can mediate the recognition and transmission of the redox signal [47,48]. The PAS1 domain complexed with FADH_2_ is extremely difficult to obtain, since FADH_2_ is easily oxidized to FAD by oxygen. As a result, the structure of NifL has not been reported in a reduced state.

### 5.3. The NifL–NifA System Regulated by GlnK

The PII protein, which is widely distributed in bacteria, archaea, and plants, functions as a signal–transduction protein in the regulation of nitrogen fixation [49]. Several genes have been verified to encode PII paralogues in proteobacteria, such as *gln*B, *gln*K, *gln*J, *gln*Y, and *gln*Z [50]. Current evidence indicates that *A. vinelandii* carries a single gene encoding a protein belonging to the PII family, designated *gln*K [40]. The GlnK protein is a PII–like protein sensing cellular nitrogen signals encoded by the *gln*K gene, which can participate in the transcriptional regulation of NifL–NifA system through its covalently modified urylation transition in *A. vinelandii* [51,52]. The *gln*K gene is often clustered and co–transcribed with *amt*B genes encoding a membrane–bound NH_4_^+^ channel (AmtB) [50]. The expression of the *gln*K–*amt*B operon in *A. vinelandii* is not affected by a fixed nitrogen supply, which is in contrast to other bacteria, such as *E. coli* and *K. pneumoniae* [53]. The GlnK protein exists in the form of a trimer structurally, which can be reversible and covalently modified by the uridylyltransferase/uridylyl–removing enzyme (UTase/UR) GlnD encoded by *gln*D, a sensor–regulator that responds to the intracellular glutamine concentration. Each GlnK trimer can be covalently modified by up to three uridine groups. In nitrogen–excess conditions, intracellular nitrogen metabolism is active, and glutamine, as an important nitrogen metabolic intermediate, is accumulated in the intracellular environment. Subsequently, GlnD can exert UR activity to catalyze the deuridine acylation of GlnK via binding to glutamine. The unmodified GlnK is involved in the GlnK–NifL–NifA ternary complex formation that can suppress NifA activity [51]. It is possible that the unmodified GlnK does not interact directly with NifA but interacts with the C–terminal GHKL domain of NifL to enhance the inhibition of NifA activity by NifL [54]. Under intracellular nitrogen–limiting conditions, the synthesis of glutamine is probably blocked, leading to a decline in glutamine concentration. The lower concentration of glutamine reverses GlnD activity, allowing it to exert UTase activity to catalyze the uridine acylation of GlnK. Consequently, the uridine acylation of GlnK induces conformational changes, abrogating its capability to interact with the NifL–NifA complex [53,55].

Mutations in GlnD that decrease its activity in the uridine adenylation of Glnk can block the synthesis of nitrogenase via stabilizing the formation of the GlnK–NifL–NifA ternary complex in *A. vinelandii* [55]. In addition, in vitro binding assays proved that unmodified GlnK can promote the formation of the NifA–NifL complex, while uridine–acylated GlnK lost such an ability. These results supported the function of GlnK in regulating the NifL–NifA complex. The previous crystal structure showed that the working mechanism of GlnK is reversibly uridylylated at the conserved Try51 by the GlnD protein in *E. coli* in the presence of low nitrogen levels. The conserved Try51 is located on the T–loop interacting with the target protein in *E. coli*, and the uridine acylation at Try51 can enhance the flexibility of the T–loop, which is not conducive to the interaction between Glnk and the target protein [56]. This result was further supported by the fact that the GlnK–Y51F mutation failed proper uridine acylation and constant inhibition of NifA activity by NifL [51]. Except for reversible modification of uridine acylation, GlnK may suffer from a non–reversible modification in direct response to nitrogen availability. The irreversible modification was identified as specific cleavage of the first three N–terminal amino acids of Glnk in *Streptomyces coelicolor*, which is speculated to be caused by ammonium shock. However, it has not yet been verified that the specific cleavage of the three N–terminal amino acids can affect the regulation of NifA activity by NifL by modulating GlnK stability [57].

The interaction of the PII protein with other proteins is also modulated by the binding of the effectors, including adenylylate energy charge (ADP and ATP) and 2–OG, which regulates signal–transduction proteins, metabolic energy, and permeases involved in nitrogen assimilation [58,59]. GlnK itself can directly perceive nitrogen limitation in response to 2–OG and ATP/ADP. Moreover, 2–OG, at the appropriate concentration, is a prerequisite for the interaction between unmodified GlnK and NifL in *A. vinelandii*, consistent with the interaction between PII proteins and other targeted proteins in *E. coli* [29,31]. One GlnK trimer in *A. vinelandii* can bind two to three 2–OG molecules, but 2–OG at high concentration (2 mmol/L) is unable to disrupt the interaction between unmodified GlnK and NifL. This suggests that GlnK is not sensitive to the concentration change of 2–OG within the physiological range in *A. vinelandii*. Therefore, the regulatory signal of GlnK uridylation is more crucial than 2–OG for GlnK– regulated NifL activity.

## 6. The Differences in the NifA–NifL System for the Different Nitrogen–Fixing Bacteria

The homologs of NifA can be identified in almost all nitrogen–fixing bacteria from proteobacteria, and most of the NifA proteins encoded by these bacteria are sensitive to oxygen or excess fixed nitrogen signals. In these microorganisms, NifA interacts with its sensor NifL protein or PII family signal–transduction proteins when fixing nitrogen. However, the regulation mechanisms are considerably varied among the different organisms.

NifA has been described to be regulated by the sensor NifL from the members in γ–proteobacteria, such as the well–studied *A. vinelandii* and *K. pneumoniae* [60,61]. In spite of the functions of NifL being similar to that in *A. vinelandii* and *K. pneumoniae*, the NifL–NifA system may adopt a different mechanism to respond to the cellular nitrogen status. In contrast to *A. vinelandii*, at least two homologs of PII–like proteins are involved in nitrogen control in *K. pneumoniae*, designated *gln*B and *gln*K. The two PII proteins are structurally similar, while the response of NifL to fixed nitrogen levels is only dependent on GlnK, encoded by *gln*K. GlnK plays a key role in relieving the inhibitory effect of NifL on NifA, but this regulation is associated with the concentration of GlnK other than the uridylation state of GlnK. GlnB encoded by *gln*B can counteract this modulation of GlnK on the NifL–NifA complex [62]. In addition, studies have shown that the NifL proteins from the two organisms differentiate. The NifL protein from *K. pneumoniae* can only be synthesized in a nitrogen–excess condition, and its C–terminal sequence has little sequence homology with the GHKL domain and none of the characteristics of binding ATP/ADP. The N–terminal PAS domain can bind and hydrolyze ATP, but only the C–terminal domain can form the stabilized complex with NifA [63]. Besides γ–proteobacteria, NifL homologs have also been characterized in α–, β–, and ζ–proteobacteria, but few studies have described the regulation of NifL in these proteobacteria [64,65].

In α– and β–proteobacteria, few species can express NifL. This suggests that the activity of NifA proteins does not require the synergistic regulation of NifL proteins in most α– and β–proteobacteria. NifA is sensitive to oxygen and fixed nitrogen signals and can directly interact with signal–transduction proteins from the PII family in response to fixed nitrogen levels [65]. Studies have shown that NifA is diversely expressed in these organisms. NifA in Rhizobias, such as *Sinorhizobium meliloti* and *Bradyrhizobium japonicum* from α– or β–proteobacteria, contains a broadly conserved cysteine–rich motif located at an inter–domain linker (IDL) region between the AAA domain and the HTH domain, which confers its oxygen sensitivity. However, the IDL region does not exist in either *A. vinelandii* or *K. pneumoniae* NifA [66]. These conserved cysteine residues are also present in many non–rhizobia NifA homologs, such as *Azospirillam brasilense* (*A. brasilense*) and *Herbaspirillum seropedicae* (*H. seropedicae*) [67]. Studies on *A. brasilense* and *H. seropedicae* have shown that four conserved cysteine residues (C414, C426, C446, and C451) make NifA sensitive to oxygen. The sensitivity of NifA to fixed nitrogen in *A. brasilense* is regulated by its two PII proteins, designated GlnB (encoded by *gln*B) and GlnZ (encoded by *gln*Z). GlnB and GlnZ are structurally similar but distinct in their regulation of nitrogen metabolism. The uridylation activity and ATP–binding ability of GlnB require NifA activation [68]. GlnB can interact with the GAF domain to activate NifA in nitrogen–limiting conditions. GlnZ may play a critical role in prohibiting the expression of the nitrogenase gene [69]. In contrast, NifA can be inhibited via interacting with the PII protein to adapt to the nitrogen–excess condition in *Rhodobacter capsulatus* [70]. These results suggest that NifA and NifL in different nitrogen–fixing bacteria may play completely different roles in regulating the expression of the nitrogenase gene. As a result, it is of great significance to further improve the regulation network of nitrogen–fixing to facilitate agricultural applications in different model strains.

## 7. The Potential Applications of the NifA–NifL System in Agricultural Development

The application of industrial nitrogen fertilizers, such as urea and nitrate, satisfies the demand for high crop yields but also generates some global “nitrogen problems” [71]. Thus, biological nitrogen fertilizers can substitute industrial nitrogen fertilizers in sustainable systems. Biological nitrogen fixation depends on functionally active nitrogenases. The complex NifL–NifA regulatory system ensures that the synthesis of functionally active nitrogenases only occurs under the proper physiological conditions. This regulation mode can ensure the normal growth of nitrogen–fixing bacteria, other than facilitating the discharge of ammonia into the environment. Thus, it is desirable for engineering methods to achieve stabilized and efficient nitrogen fixation that provides an alternative to synthetic nitrogen. In recent decades, various approaches have been proposed to enhance the efficiency of biological nitrogen fixation by offering synthetic nitrogen alternatives (Figure 5). Some of them have been successfully applied in agriculture to achieve high crop yields.

Studies have shown that the application of nitrogen–fixing bacteria can stimulate the growth of crops, and the direct inoculation of azotobacter can improve the yield of crops such as cereal, potatoes, corn, and vegetables. [8]. In nature, most azotobacter are autotrophic heterotrophic bacteria, which can fix about 20 kg of nitrogen per hectare per year. In a farmland ecosystem, most of the fixed nitrogen can be utilized in crop production as a substitute for a portion of nitrogen fertilizer. Although nitrogen–fixing bacteria can be directly introduced into agricultural production as biological nitrogen fertilizer to improve crop yields, the nitrogen–fixation efficiency of nitrogen–fixing bacteria is still insufficient to meet the demand of agricultural development for nitrogen fixation. Researchers believe that these biotechnological approaches, which allow crops to make their own nitrogen fertilizers, engineer legume symbiosis into cereals, and genetically manipulate azotrophic bacteria for efficient nitrogen fixation and ammonia excretion, with the potential for even greater returns. Nearly half a century ago, Ray Dixon and his colleagues managed to transfer a complete nif gene cluster from *K. pneumoniae* to *E. coli* to confer the ability of nitrogen fixation, which laid the groundwork for its application to crops [72]. This technique has been successfully applied to tomatoes as well, which enabled the plant to fix nitrogen and increase its yield. Moreover, the technology is now being applied to wheat, corn, and other large–scale applied experiments [1]. Despite some amazing breakthroughs in the ability of plants to fix nitrogen, several technical obstacles remain to be overcome. It is the basic prerequisite for engineering nitrogen fixation that the nitrogenase system is expressed stably and intactly in plant cells. In addition, protecting nitrogenase from oxygen is a key barrier since nitrogenase is sensitive to oxygen, as described previously, and plant cells generate large amounts of oxygen. Recent work on the transfer of nitrogenase into plant cells has shown that mitochondria and chloroplasts can be used as suitable locations for nitrogenase expression, as they can provide a large amount of reducing agents and ATP for the nitrogen–fixation process [73,74,75,76,77,78]. In addition, the aerobic respiration of mitochondria creates an oxygen–depleted environment for the expression of active nitrogenase, similar to that of aerobic azide–trophic bacteria, such as *A. vinelandii* [73,79,80]. In fact, the direct introduction of nitrogenase into plant cells can confer crops to make their own nitrogen fertilizer, but the plant cells cannot express the components of the regulatory system, leading to an out–of–control NifL–NifA regulatory system. The process of nitrogen fixation requires a large amount of ATP and reducing agents, so extra–fast or excessive nitrogen fixation may cause irreversible damage to plant cells and lower the yield of crops. TCA in eukaryotic cells is carried out in the mitochondrial matrix, which provides multiple regulators for the NifL–NifA regulatory system, including 2–OG, FAD/FADH_2_, ATP/ADP, and Glu/Gln [81,82]. Studies on the subject suggest that mitochondria may be used as the working site for the NifL–NifA regulatory system. Thus far, no successful application of the NifL–NifA regulatory system in plant cells has been reported.

Even if such work was technically successful, it is difficult for such strategies to gain public approval because they involve genetic modification. Therefore, focusing on the nitrogen–fixing bacteria themselves, and adjusting the nitrogen–fixing patterns of the nitrogen–fixing bacteria to generate more fertilizer, may achieve higher rewards. Studies have shown that diazotrophic bacteria can be engineered for excess production and excretion of NH_3_ using several strategies [12,83,84]. Precise disruption of the *nif*L component in the *nif*LA operon results in phenotypic dysregulation, the production of ammonium far exceeding intracellular requirements, and the release of up to 30 mM of ammonium into the growth medium [39,60,85,86]. However, in the absence of active NifL, the activity of NifA becomes uncontrollable, which results in the generated bacterial strains being disadvantaged in the environment due to the energetic burden [85,86]. This suggests that it is unrealistic to engineer NifL to remove nitrogen regulation. Researchers believe that both excessive ammonia emission and the minimization of the energy burden can be achieved by controlling the activity or activation mode of NifA in a nitrogen–limiting environment [36,83]. In a nitrogen–limiting environment, the NifA–E356K mutant in engineering *A. vinelandii* is regulated by the 2–OG level instead of NifL [36]. After the key residues of NifA in other proteobacteria are replaced by that of NifA in *A. vinelandii*, this variant exhibits analogous properties to that of NifA in *A. vinelandii*. This may contribute to the engineering of carbon–source–dependent ammonia excretion, which is distinct among the different members of this family [83]. In addition, PII proteins, such as GlnK, modified by UTase/UR, can directly or indirectly control NifA activity in response to fixed nitrogen signals [51,54]. Based on this mechanism, the engineering of fixed–nitrogen bacteria for insensitivity to the intracellular nitrogen state, to continuously fix nitrogen and exhaust ammonia, has been accomplished by mutating the PII protein in *Azorhizobium caulinodans* (*A. caulinodans*) and *A. brasilense* [87,88,89]. However, this engineering strategy does not seem to be universally applicable to other bacteria and is even fatal, limited by differentiated functions of PII in the regulation of NifA and nitrogenase activity [90,91,92].

From the perspective of agricultural development, engineering diazotrophic bacteria for excessive nitrogen fixation and ammonia excretion may cause two major detriments. First, uncontrolled NifA activity and active nitrogenase expression impose a severe energetic burden and eliminate the competitiveness of root colonization. Second, ammonia excretion is not directed, which results in low utilization of ammonia in the target crop. This is expected to establish symbiotic engineering diazotrophic bacteria that can target crops for ammonia excretion, similar to the symbiotic relationship between soybean and rhizobia. Poole and colleagues developed a symbiotic model in which an engineered diazotrophic bacterium, the endophyte *A. caulinodans*, sensed only the nitrogen–fixing synthetic rhizopine signaling released by barley, which could induce ammonia excretion in the barley direction [93,94,95]. Subsequently, this technology was successfully established on the symbiotic model between *A. caulinodans* and cereals [96]. This highly complex control circuit represents an important milestone in the development of “synthetic symbiosis” in which N_2_ fixation and NH_3_ excretion can be activated in bacteria–specific colonizing target rhizopine–producing cereals, targeting the delivery of nitrogen to the crops while avoiding potential interactions with non–target plants.

## 8. Summary and Prospect

The “Nitrogen problem” restricts the sustainable development of agriculture. The application of the biotechnology approach to the fabrication of biological nitrogen fertilizer presents an opportunity to address this issue [97]. Currently available biological nitrogen fertilizers, either wild–type or engineered, cannot match the ability of chemically synthesized fertilizers to improve crop yields. However, it cannot be denied that biological nitrogen fertilizers have great potential in agricultural applications.

*A. vinelandii* is widely used because of its high efficiency of nitrogen fixation and simple genetic material that make it an easy subject for genetic manipulation, as well as its agricultural value being gradually embodied. Several engineered bacteria have been designed and developed with respect to *A. vinelandii* that promise a wide range of applications in agriculture. However, a significant amount of work has to be completed on agricultural applications of *A. vinelandii*. Improving the value of agricultural applications depends on the development of new biotechnological data. To advance and develop new biotechnological applications and products, a better understanding of their nitrogen–fixation and –regulation mechanisms is required. Although the nitrogen–fixing regulatory process of *A. vinelandii* is becoming clearer, an accurate explanation of the molecular mechanism is still missing due to a lack of relevant structural biological evidence. For example, how does 2–OG mediate the allosteric of GAF domain to activate NifA? When NifA activates transcriptional activity of the RNAP–σ54–DNA holoenzyme, what is the molecular mechanism by which an L1/L2 loop binds to σ54 and leads to partial uncoupling of the DNA double strand? What is the molecular mechanism by which NifL inhibits NifA activity under the conditions of excess nitrogen? In addition, structural information on how unmodified GlnK forms ternary complexes with NifL and NifA is still unclear. These questions need to be answered by reliable biochemical and structural biological data theoretically supporting the artificial design and efficient utilization of nitrogen–fixation modules.

## Figures and Tables

**Figure 1 ijms-24-00907-f001:**
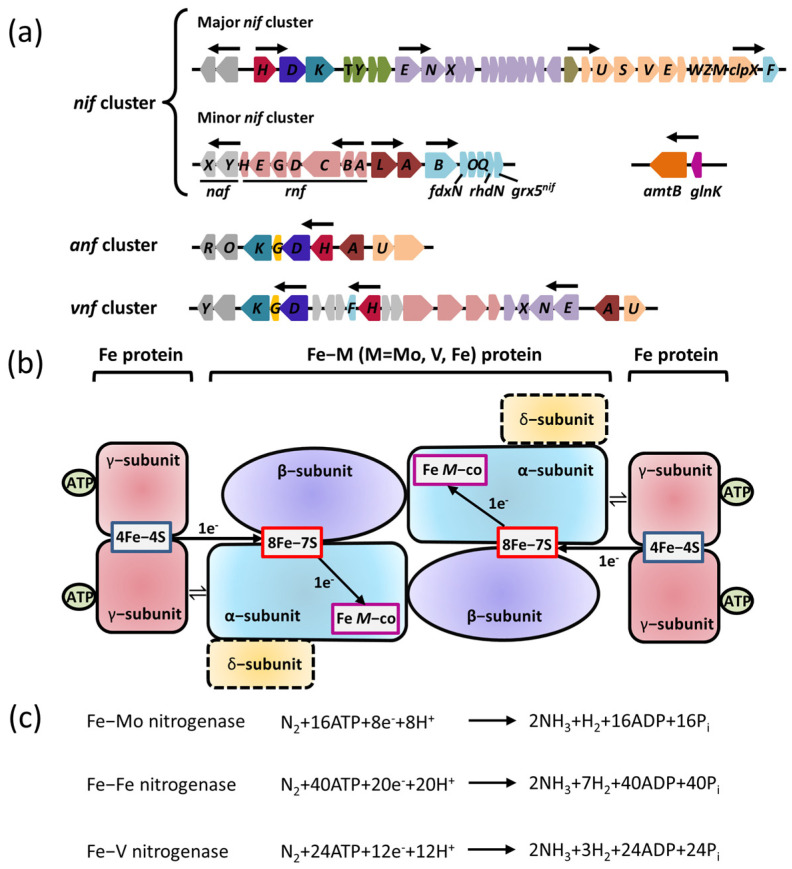
Genes encoding the three forms of nitrogenases and their regulatory proteins are required for nitrogen fixation. (**a**) The organizations of *nif*, *anf*, and *vnf* clusters encoding the three forms of nitrogenases in *A. vinelandii*. Predicted σ54–dependent promoter regions are depicted by arrows. Shown are genes encoding the regulatory proteins with known or predicted functions (dark brown), the components involved in the catalytic reduction of N_2_ (*nif*D in blue; *nif*K in aqua; *nif*H in wine red; *nif*G in yellow), the assembly or stability of nitrogenase (orange), the maturation of nitrogenase (olive), the maturation of Fe–M cofactor (purple), electron transfer (light red), the biosynthesis of Fe–M cofactor (light aqua), and an unknown function (gray). (**b**) A diagram of the three forms of nitrogenases involved in electron transfer. M is Mo, Fe, or V. The α–subunit is encoded by *nif*K; β–subunit is encoded by *nif*K; γ–subunit is encoded by *nif*H; δ–subunit is encoded by *nif*G. (**c**) The nitrogen–fixation reaction catalyzed by the three forms of nitrogenases.

**Figure 2 ijms-24-00907-f002:**
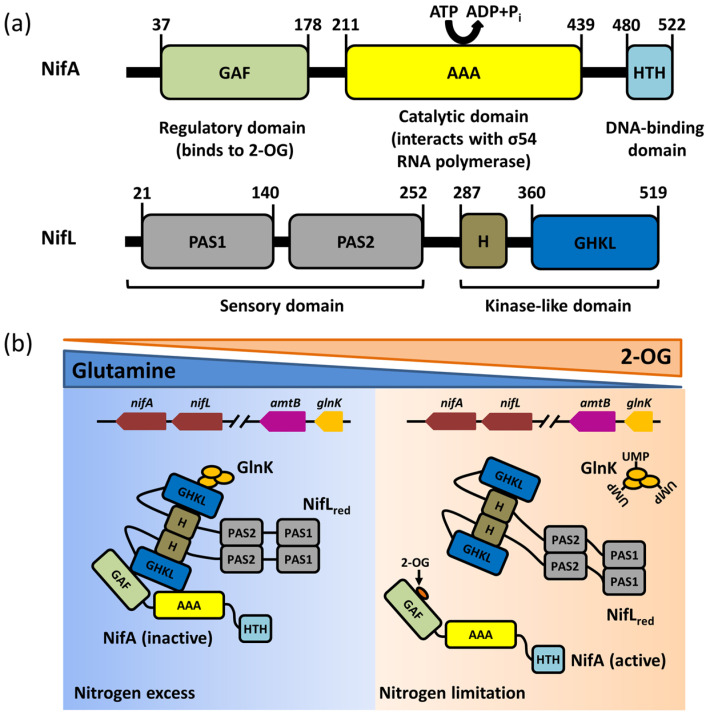
The domain structures and functions of NifA and NifL. (**a**) Schemes of the domain structures of the NifA and NifL protein; (**b**) The distinct regulatory modes of NifA–NifL–GlnK complex in the excess or limitation of nitrogen. In nitrogen excess, GHKL domain of NifL protein interacts with NifA protein to inhibit NifA. Glnk can interact with NifL, enhancing the inhibition of NifA activity by NifL. When nitrogen is limited, the uridylylation of Glnk disrupts the interaction between Glnk and NifL. 2–OG can bind to GAF domain of NifA to remove the inhibition of NifA activity by NifL.

**Figure 3 ijms-24-00907-f003:**
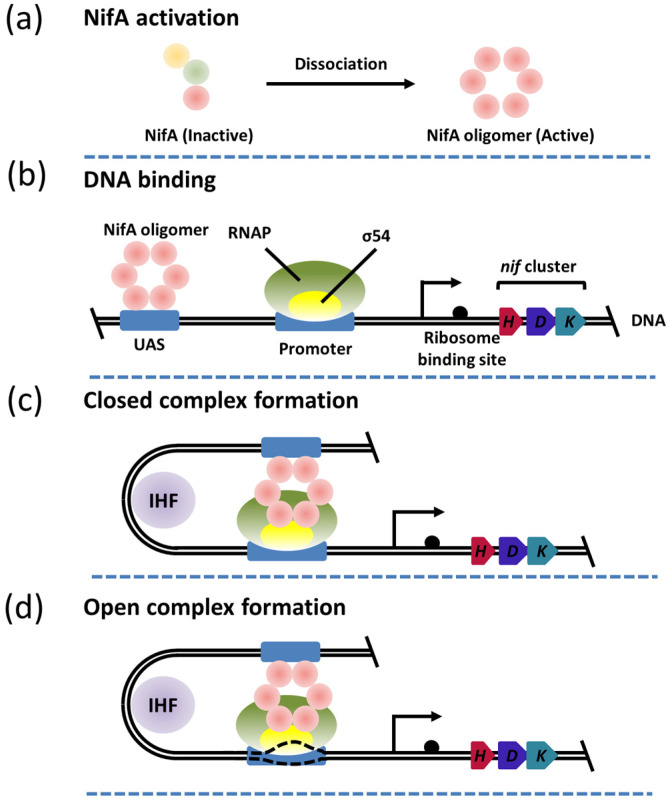
The transcriptional initiation process mediated by *nif* cluster promoted by NifA. (**a**) NifA activation. Active NifA needs to be reconstituted into an oligomeric state (predicted as hexamer). The pink ball represents NifA, and green ball represents NifL. The Glnk is represented by brown ball. (**b**) DNA binding. Oligomeric NifA can bind to the UAS region; (**c**) The formation of the closed complex. IHF can bend DNA to promote the interaction of oligomeric NifA with RNAP–σ54; (**d**) Open complex formation. Oligomeric NifA provides energy for DNA opening promoted by RNAP–σ54.

**Figure 4 ijms-24-00907-f004:**
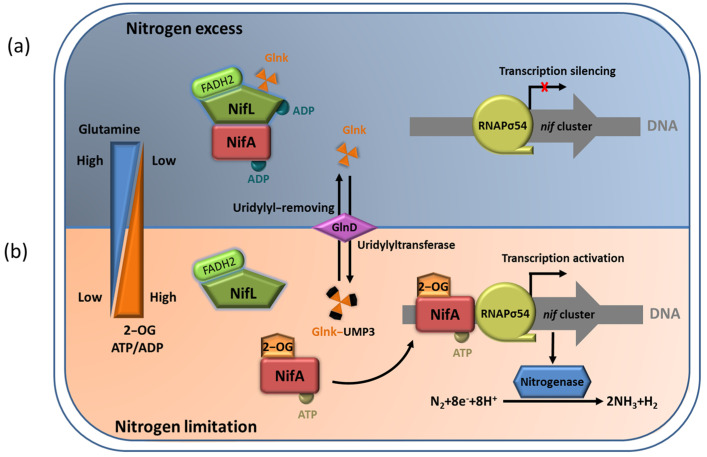
The NifL–NifA system’s responses to environmental and metabolic conditions in *A. vinelandii*. (**a**) In nitrogen–excess conditions, the binding of high concentration of glutamine to GlnD results in uridylyl removal from Glnk. NifL forms the binary complexes with NifA to inhibit NifA activation. In addition, the non–covalently modified GlnK can also interact with NifL to promote the formation of the GlnK–NifL–NifA ternary complex and inhibit NifA activity, leading to the transcriptional silencing of *nif* cluster; (**b**) In nitrogen–limiting conditions, high concentration of 2–OG can result in the association of NifL and GlnK in the NifA–NifL–Glnk complex. Free NifA can activate the transcription of nitrogenase gene.

**Figure 5 ijms-24-00907-f005:**
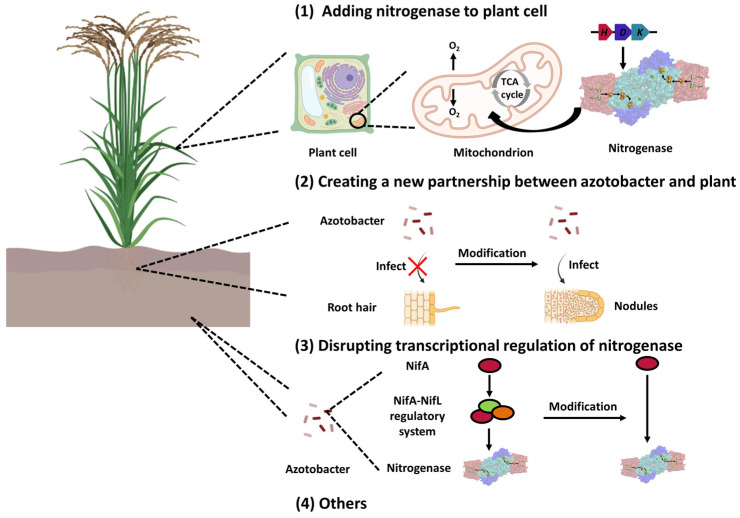
The strategy of engineering nitrogen fixation. (**1**) Adding nitrogenase into plant cells. Nitrogenase requires a lot of energy and a low–oxygen environment to execute the nitrogen–fixing function. The mitochondrion provides a suitable place to accommodate the enzyme. (**2**) Creating a novel partnership between azotobacter and plants. Under natural conditions, non–symbiotic azotobacter are not capable of infecting crops. Several key modifications, such as expressing rhizopine in the crop cells and the receptor of rhizopine in azotobacter, can specifically allow crops to recognize azotobacter. The elongated root hair can create a tunnel for azotobacter to infect the root hair and form nodules. (**3**) Disrupting the transcriptional regulation of nitrogenase. As the transcriptional regulation of nitrogenase is the crucial rate–limiting step in nitrogen fixation, several modifications can disrupt the original regulatory pathways, allowing them to evolve in ways beneficial to agricultural development. (**4**) Other strategies.

## Data Availability

Not applicable.

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
