# Peer review of "Molecular Mechanism and Agricultural Application of the NifA–NifL System for Nitrogen Fixation"

_ijms, 2023, doi:10.3390/ijms24020907_

Round 1

Reviewer 1 Report

The present work summarizes the current knowledge on the nitrogenase system of the bacterium Azobacter vinelandii. The study is exhaustive and well documented, although at times it is somewhat dense. Perhaps the last part, which describes the real potential for field application, is a bit brief compared to the length dedicated to the regulatory systems.

Minor corrections:

1. L. 85: Replace Fe-V nitrogenase with Fe-Fe nitrogenase.

2. L. 445-446: "Unlike A. vinelandii, at least two homologs of the PII-like protein are involved in nitrogen control...". Change the verb to the plural form.

3. L. 582: Since this is the first time it is named, include the full name of the bacterium A. caulinodans.

4. All scientific names should be in italics.

Author Response

The present work summarizes the current knowledge on the nitrogenase system of the bacterium Azobacter vinelandii. The study is exhaustive and well documented, although at times it is somewhat dense. Perhaps the last part, which describes the real potential for field application, is a bit brief compared to the length dedicated to the regulatory systems.

Response: We are very grateful for the reviewer's affirmation. It is a developing study that the mechanism of regulating nitrogen fixation is applied to agriculture. We have tried to collect more studies to surpport the real potential for field application, while few more studies are relevant to our content.

Minor corrections:

Point 1:    L. 85: Replace Fe-V nitrogenase with Fe-Fe nitrogenase.

Response 1: We apologize for the confusion. As reviewer suggested, “Fe-V nitrogenase” had been replaced by “Fe-Fe nitrogenase” in line 85.

Point 2: L. 445-446: "Unlike A. vinelandii, at least two homologs of the PII-like protein are involved in nitrogen control...". Change the verb to the plural form.

Response 2: We apologize for the confusion. We have revised the plural form in line 445-446. This also alerted us. Therefore, we have carefully examined all plural form in the manuscript and made many revisions.

Point 3: L. 582: Since this is the first time it is named, include the full name of the bacterium A. caulinodans.

Response: We thank the reviewer for the constructive suggestion. As reviewer suggested, we have carefully examined the whole manuscript and found that “A. caulinodans” have first named in line 570 rather than line 582. Therefore, we have executed the full name in line 570. Furthermore, we have also examined all bacterium names and executed the full name of the bacterium in first time.

Point 4: All scientific names should be in italics.

Response: We thank the reviewer for the constructive suggestion. We have changed all scientific names in italics.

Reviewer 2 Report

the authors address a valid problem for agriculture. 

However the manuscript is so difficult to read because of the English-  please please seek professional editing. Also the organization of sections could be improved. Address all the abbreviations used -  some were not explained in paper.  And the legends for figures require more detail to aid understanding 

Author Response

The authors address a valid problem for agriculture. However the manuscript is so difficult to read because of the English-  please please seek professional editing. Also the organization of sections could be improved. Address all the abbreviations used some were not explained in paper.  And the legends for figures require more detail to aid understanding.

Response: Thank you for your comments. According to your suggestion, English editing was performed for the full revised manuscript by using an English language editing service (MDPI). In addition, we corrected all the misspellings or errors that we found, and improved the organization of all sections. We also replenished the missing abbreviations, and filled in the figures legends and texts in more detail.

Corrections:

Point 1: L.29-L.30 urea or Nitrate ions etc.?

Response 1: Thank you for your comments. This is not the case. The inorganic nitrogen that plants can absorb and utilize is mainly NO3- and NH4+ (PMID: 34537962). Here, we want to express the view in nature that most of the inert nitrogen (N2) needs to be converted into ammonia by microbial nitrogen fixation. The ammonia can be converted by nitrobacteria to nitrate that are more easily absorbed by plants. The most of urea used in agricultural production comes from industrial nitrogen fixation.

Point 2: L.35-L.37 nitrogen fertilizer run off into water systems is major   so redo

Response 2: We thank you for the constructive suggestion. According to your suggestion, we rewrote this part.

Point 3: L.52-L.58 much more than just N and P,  K Fe Si etc

Response: Thank you for your comments. According to your suggestion, we deleted this word.

Point 4: L.58 need reference

Response: Thank you for your comments. According to your suggestion, we cited three articles as the reference 9-11.

Point 5: L.120 Figure 1 this image is fuzzy, needs improvement

Response: Thank you for your comments. According to your suggestion, we have modified Figure 1.

Point 6: L.131 genes  when more than one respectively

Response: Thank you for your comments. According to your suggestion, we have checked and fixed all the plurals of nouns.

Point 7: L.165-L.166 diagram?

Response: Thank you for your comments. We are sorry to confuse you. According to your suggestion, we modified the content of this part and added the diagram at the end of this word.

Point 8: L.170-L.173  this needs more explanation and figure???

Response: Thank you for your comments. According to your suggestion, we added the explanation and relative figures to the manuscript.

Point 9: L.197 where is nitrogen excess explained in full.

Response: Thank you for your comments. According to your suggestion, we added the explanation for nitrogen excess and nitrogen limitation in the end of part 2.2 and Figure 2 legend.

Point 10: L.204-L.205 NifA-like homolog 1 (Nlh1), a NifA homolog?

Response: Thank you for your comments. According to your suggestion, we have deleted the pharase“a NifA homolog”.

Point 11: L.226 In Figure 3(a), what are the other two blobs in inactive NifA?

Response: Thank you for your comments. We are sorry to confuse you. In Figure 2(b), we introduced the formation of ternary complex (GlnK-NifA-NifL) in nitrogen excess. In this figure legend, we have noted GlnK used in yellow blob and NifL used in green blob.

Point 12: L.321   italics   this is not a  N fixer  so generalized response?

Response: Thank you for your comments. We are sorry to confuse you. According to your suggestion, we have modified “Escherichia coli” with “E. coli”. In addition, the concentration of 2-OG reflects the level of carbon and nitrogen metabolism, which is generalized response. E. coli itself has not the function fixing nitrogen due to lacking of nitrogenase, thus it is not a N fixer.

Point 13: L.355 alternative to what?

Response: We apologize for the confusion. After our discussion and document research, we think that “alternative signal” doesn’t seem to be suitable for using in here. Therefore, we have modified this word in the revised version.

Point 14: L.374 “The role of PII protein functions as a signal transduction protein in regulation of nitrogen fixation is widely distributed in bacteria, archaea and plants”?

Response: We apologize for the confusion. We have replaced “The role of PII protein functions as a signal transduction protein in regulation of nitrogen fixation is widely distributed in bacteria, archaea and plants” with “PII protein that is widely distributed in bacteria, archaea and plants functions as a signal transduction protein in regulation of nitrogen fixation” in this manuscript.

Point 15: L.434 why is this section here, as NifA was a key player in earlier sections?

Response: The molecular mechanism of NifA-NifL system may be different in different nitrogen-fixing bacteria. Azotobacter vinelandii is a well-established model bacterium for studying the mechanism of nitrogen fixation. The study on NifA-NifL system in Azotobacter vinelandii is relatively perfect compared with other nitrogen-fixing bacteria. Therefore, in earlier sections, we have systematically summarized the molecular mechanism of NifA-NifL system in Azotobacter vinelandii. In this section, we have briefly summarized the information of NifA-NifL system that has been reported in other nitrogen-fixing bacteria species. We can have a deeper understanding of the NifA-NifL systems via comparing the differences of NifA-NifL systems between Azotobacter vinelandii and other nitrogen-fixing bacteria species. In addition, we think that the subtitle is not appropriate in this section. Thereforce, we have modified the subtitle with “The differences of NifA-NifL system among the different nitrogen-fixing bacteria”.

Point 16: L.460 but next sentence is about NifA  do not understand why arranged this way.

Response: We apologize for the confusion. This is due to our negligence, which led to the wrong deletion of part of the content. We have supplemented the information with “This suggests that the activity of NifA proteins does not require the synergistic regulation of NifL proteins in most of the α-and β-Proteobacteria.” in the revised manuscript.

Point 17: L.484 define what you mean, nitrate  ammonia  urea?

Response: We thank the reviewer for the constructive suggestion. As reviewer suggestion, we have defined industrial nitrogen fertilizer in the revised manuscript.

Point 18: L.497 what if azotobacter does not colonize other crop plants well enough?

Response: It is a good strategy for promoting the utilization efficiency of biological nitrogen fixation that azotobacter can colonize in crop plants via simulating the symbiotic relationship between legumes and rhizobia. The symbiotic relationship between legumes and rhizobia mainly relate to the rhizopine signal, such as SI (scyllo-inosamin), synthesized by legumes and the signal perception of rhizopine signal by rhizobia. The perception of rhizobia for rhizopine signal depends on MocR protein expressed by rhizobia. Based on these analyses, Professor Philip Poole and his colleagues have successfully they successfully constructed barley plants that could stably express SI via engineering plant (PMID: 35412890). Under the control of NifA, SI signal can promotes nodulation of rhizobia and forms symbiotic nitrogen fixation. In addition, IntABC (An ATP-binding cassette (ABC) transporter) proteins and MocB proteins are the transporter of SI and can enhance the perception of SI signal by rhizobia. Therefore, the strategies include that engineering the crop plants can stably express rhizopine signal, and engineering azotobacter can accept the rhizopine signal, which maybe efficiently build the relationship azotobacter colonizing in crop plants. The overexpression of IntABC and MocB proteins in azotobacter can enhance the colonizing ability of azotobacter.

Point 19: L.499-L.500 why low-oxygen is there?

Response: The mitochondrion is the main site of aerobic respiration in plant cells. The mitochondria, the rice-shaped organelles that crank out the cellular fuel adenosine triphosphate (ATP). Making ATP consumes oxygen, so mitochondria can be relatively anaerobic in places (PMID: 27634521).

Point 20:  L.501-L.503 define term infection?

Response: We thank the reviewer for the constructive suggestion. We have defined the term infection in figure legend of Figure 5.

Point 21: L.546-L.547 public acceptance of GEMs too a big problem

Response: We thank the reviewer for the constructive suggestion. We also think that “public approvement” should include the public acceptance of GEMs and people.

Point 22: L591  ony one of the main ones; nitrate seems a current fix; need to change attitude

Response: We thank the reviewer for the constructive suggestion. We modified the word in the the revised manuscript. We agree with the view from reviewer that nitrate seems a current fix, but it is not applicable in this manuscript. Crops have the ability to acquire nitrogen from the atmospheric N2, through nitrogen fixing bacteria, directly from assimilating inorganic nitrogen forms (nitrate or ammonium). Nitrate ingested by crops comes from two main sources: industrial nitrogen fixation and nitrobacteria. Industrial nitrogen fixation is the main cause of “nitrogen problems”. The nitration reaction executed by nitrobacteria can promote the recycling and use of bioavailable nitrogen. Therefore, enhancing the nitration reaction may improve “nitrogen problem”. However, most of the bioavailable nitrogen is metabolized and converted into N2 back into the atmosphere, and biological nitrogen fixation is also essential for improving “nitrogen problem”. In this manuscript, we have analyzed the importance of the NifA-NifL system for nitrogen fixation in agricultural applications. The direct product of nitrogen fixation is ammonium rather than nitrate. On one hand, ammonia produced by nitrogen fixation can be directly used by crops; on the other hand, ammonia is further converted to nitrate by nitrobacteria. Therefore, we think that “nitrate seems a current fix” is not applicable in this manuscript.

Round 2

Reviewer 2 Report

The authors present a full review.

Please though check the legend and discussion on azotobacter Fig 5 

it reads as if azotobacter  form nodules  to my knowledge they do not

but they can modulate nodulation by Rhizobium spp    is that what you are trying to say?   please check  

Author Response

Response to Reviewer 2 Comments

The authors present a full review. Please though check the legend and discussion on azotobacter Fig 5 it reads as if azotobacter  form nodules  to my knowledge they do not but they can modulate nodulation by Rhizobium spp    is that what you are trying to say?   please check 

Response: We thank the reviewer for the constructive suggestion. As reviewer suggestion, we carefully have checked the legend and discussion on azotobacter Fig 5. We are trying to say that modified azotobacter may form nodules in root hair. As we explain in the legend, non-symbiotic azotobacter cannot form nodules in nature. Rhizopine signals produced by legume plants can promote nodule formation by rhizobia in legume roots. The perception of rhizobia for rhizopine signal depends on MocR protein expressed by rhizobia. If we can construct efficient perception of rhizine signals in azotobacter, azotobacter may form nodules and colonize in crop plants via simulating the symbiotic relationship between legumes and rhizobia.

Corrections:

Point 1:    L.168-L.171 expression? do you mean activity? we generally use expression for gene activation.

Response 1: We apologize for the confusion. In this section, “expression” is the expression of nitrogenase gene. “activity” is the activity of NifA protein. The NifA protein is not equivalent to nitrogenase and only used as an activator for transcriptional initiation of nitrogenase gene. The expression of nitrogenase gene is relative with the activity of NifA. When there is excess nitrogen, low concentrations of 2-OG is not enough to activate NifA. Inactive NifA have not the function promoting the transcription of nitrogenase. Therefore, the expression of nitrogenase is limited. When nitrogen is limited, high concentrations of 2-OG can activate NifA. Active NifA protein can function as an activator for the transcription initiation of nitrogenase gene. In brief, activation of NifA protein is a necessary condition for the expression of nitrogenase.

Point 2: L.231  what is reseda    rose?  red? pink?

Response 2: We apologize for the confusion. “reseda” had been replaced by “green”.

Point 3: L.490-L.491  can?   in sustainable systems

Response: We thank the reviewer for the constructive suggestion. As reviewer suggestion, we modify this section.

Point 4: L.505 plants?

Response: We apologize for the confusion. As reviewer suggestion, we modify this section.

Point 5:    L.506 azotobacter can gain ingress but to my knowledge do not produce the nodule structure      so unsure of these statements   do you mean other types of N fixing bacteria for this section?

Response: We thank the reviewer for the constructive suggestion. According to our investigation, more and more researchers focus on getting plants, such as cereal, barely and corn, to exchange signals with nitrogen-fixing bacteria, which can help nitrogen-fixing bacteria to invade plant roots, where these bacteria release ammonia in exchange for carbohydrates made by plants. This good relationship contributes to the high yield of crops. In nature, azotobacter cannot produce the nodule structure because of lacking of the exchange signals between azotobacter and plants. After several modifications, it establishes the exchange signals between azotobacter and plants, and large numbers of nitrogen-fixing bacteria can colonize in the roots of plants. Then the plants need to be tweaked to form root nodules and to make their bacterial guests comfortable and productive (PMID: 27634521).

Point 6: L.507 production of rhizopine in the plant host and a receptor for rhizopine in the azobacter are required for the productive rhizobacter -plant association.

Response: According to our investigation, production of rhizopine in the plant host and a receptor for rhizopine in the azobacter are required for the productive rhizobacter-plants association. The rhizopine produced by the plant host can use as a signal to recruit nitrogen-fixing bacteria to colonize in plant roots. In addition, rhizopine can promote nodule formation by azotobacter in plant roots.

Point 7:    L.509 do not think this is true  need references specific to azotobacter

Response: In 2016, Erik Stokstad has carefully introduced the two routes to self-fertilizing plants in the review (PMID: 27634521). Thereinto, he has summarized the strategy creating a novel partnership between nitrogen-fixing bacteria and plants. In addition, he also briefly described the efforts and achievements made by different teams in the research of this strategy, and predicted that this strategy is feasible but there is a long way to go. Azotobacter is nitrogen-fixing bacteria. According to our investigation, we have no found that any azotobacter have successfully applied this strategy. The view that modified azotobacter can form nodules in the root hair of plants is inferred from the conclusion of Erik Stokstad. Based on the strategy creating a novel partnership between nitrogen-fixing bacteria and plants, Professor Philip Poole and his colleagues have successfully constructed barley plants that could stably express rhizopine signal via engineering plant. In addition, they have modified A.caulinodans to increase the sensitivity of A.caulinodans to the rhizopine signal. Using this improved genetic circuitry, they have successfully detected that modified A.caulinodans can colonize in the root of barley plants (PMID: 35412890). This work represents a key milestone toward the development of a synthetic plant-controlled symbiosis and can provide some reference for creating the partnership between azotobacter and plants.

Point 8: L.566  what is meant?

Response: We apologize for the confusion. “In an entry-limiting environment” has been replaced by “In a nitrogen-limiting environment”.

Point 9: L.590  expand this??

Response: The symbiotic model is not to expand ammonia excretion, but to direct ammonia excretion for barley.